# Super stretchable hydrogel achieved by non-aggregated spherulites with diameters <5 nm

Guoxing Sun[1], Zongjin Li[1], Rui Liang[1], Lu-Tao Weng[2,3] & Lina Zhang[4]

The scope of hydrogel applications can be greatly expanded by the improvement of mechanical properties. However, enhancement of nanocomposite hydrogels (NC gels) has been severely limited because the size of crosslinking nanoparticles is too large, at least in one dimension. Here we report a new strategy to synthesize non-aggregated spherulite nanoparticles, with diameters <5 nm, in aqueous solution, and their enhancement to hydrogel. The stress and stretch ratio at rupture of our NC gel are 430 and 121 KPa with only 40-p.p.m. nanoparticle content. The NC gel containing 200-p.p.m. nanoparticles can revert to 90% of its original size after enduring 100-MPa compressive stress. Our results demonstrate that the suppression of nanoparticle size without aggregation helps to establish a super stretchable and high-toughness hydrogel network at very low inorganic content.

[1] Department of Civil and Environmental Engineering, The Hong Kong University of Science and Technology, Clear Water Bay, Hong Kong 999077, China. [2] Materials Characterization and Preparation Facility, The Hong Kong University of Science and Technology, Clear Water Bay, Hong Kong 999077, China. [3] Department of Chemical and Biomolecular Engineering, The Hong Kong University of Science and Technology, Clear Water Bay, Hong Kong 999077, China. [4] School of Material Science and Engineering, University of Jinan, 336, West Road of Nan Xinzhuang, Jinan 250022, Shandong, China. Correspondence and requests for materials should be addressed to Z.L. (email: zongjin@ust.hk).

The mixing of nanoparticles with polymers to form composite materials has been practiced for decades. Nanoscale particles usually aggregate, and the aggregation becomes more serious with the decrease of particle size, and negates any benefits associated with the nanoscopic dimension[1]. There is a critical need, especially for nanocomposite hydrogels (NC gels), for non-aggregated nanoparticles at single-digit nanometre scale in an aqueous solution. Hydrogels can be used as scaffolds for tissue engineering[2], temporary supports for cells[3] and vehicles for drug delivery systems[4]. Conventional polymeric hydrogels usually exhibit weak strength and low toughness, and NC gels have been developed to improve the mechanical properties. In NC gels, polymer chains crosslink on the surface of the nanoparticles to establish a strong network, thus the mechanical properties of the hydrogel can be enhanced. However, either a large surface area or aggregation of the nanoparticles leads to uneven dispersion of the crosslink points, which greatly limits the property enhancement of the NC gels. In literature, the sizes of reported crosslinking nanoparticles were unfortunately all $>60$ nm at least in one dimension[5–18]. NC gel using inorganic clay particles was first reported in 2002 (ref. 5). Afterwards, various forms of sheet nanoparticles such as hectorite[6–9], montmorillonite[10], graphene oxide (GO)[11–15], layered double hydroxide (LDH)[16], titanate(IV) nanosheet[17] and controllable activated nanogels[18] were introduced to NC gels. With sizes of $>800$ nm in at least one dimension, clay has a concentration threshold (at least 1 wt%) for improving the strength of NC gels[5–9]. Such a high content of nanoparticles in the NC gels leads to residual and transparency problems. The incorporation of GO, which leads to the NC gels black and less translucent, makes the situation even worse[11–15]. Furthermore, the number of crosslink points in all these NC gels are restricted in the largest functional area ($>2,800$ nm$^2$; Supplementary Table 1) of the particles, resulting in the aggregation of the crosslink points within the NC gel, which greatly limits the enhancement of mechanical properties. As a result, high stress, stretch ratio and recoverability are difficult to achieve at the same time in a typical NC gel. For example, hydrogel enhanced by exfoliated montmorillonite layers can be stretched up to 118 times of the initial length, but the maximum stress is limited to 100 KPa (ref. 10); and for hydrogels with a stress $>200$ KPa enhanced by high-content clay nanoparticles, the stretch ratio at rupture is $<10$ (ref. 6).

A more homogeneous distribution of high-density crosslink points is expected if the isotropic size of crosslinker nanoparticles can be suppressed to single-digit nanometres, without aggregation. In this work, we develop a new strategy to prepare calcium hydroxide (Ca(OH)$_2$) nano-spherulites (CNS), with diameters $<5$ nm, which act as the crosslinker particles in polyacrylamide (PAM) hydrogel. The CNS separate individually without any aggregation, even after drying up or being implanted into the polymer matrix. The new PAM/CNS NC hydrogel system possesses many advantages over other NC hydrogels reported so far. For example, the PAM/CNS NC gel containing only 40-p.p.m. CNS has a swelling ratio of 250, a maximum stretch ratio of 121, and a stress up to 430 KPa, which are $\sim 5$, 14 and 4 times higher than those of poly(N-isopropylacrylamide)/clay NC gel with 3 wt% clay[5].

## Results

**Fabrication of CNS and PAM/CNS NC gels**. It has been reported that CNS of various sizes and small aggregates could be obtained in diols and water/oil microemulsions[19,20]. In this work, we used the hydration process of tricalcium silicate (Ca$_3$SiO$_5$), a main component of portland cement, to produce the CNS. When Ca$_3$SiO$_5$ particles are dispersed in water, calcium cations (Ca$^{2+}$) are released from the Ca$_3$SiO$_5$ and form Ca(OH)$_2$ crystals in the aqueous solution[21]. The hydrolysis temperature is critical in this process. We found that 0 °C was the optimal temperature because at this temperature, the releasing speed of calcium cations from the Ca$_3$SiO$_5$ was just enough to form Ca(OH)$_2$ spherulites and at the same time the size of the spherulites was suppressed due to the low crystallization temperature. This can be clearly seen in Fig. 1a, which shows a transmission electron microscopy photograph taken on the surroundings of a hydrolysed Ca$_3$SiO$_5$ particle. The left side of Fig. 1a corresponds to a large block of Ca$_3$SiO$_5$ with a height of 500 nm, while the right side and the inset of Fig. 1a show the tiny spherulites with diameters 5 nm. Energy dispersive spectroscopy analysis confirms that these spherulites contain only Ca and O elements, while the electron diffraction pattern (Fig. 1b) and the observed lattice structure in the inset of Fig. 1a indicate that they are crystalline CNS. In aqueous media with a concentration of 200 p.p.m., CNS have a zeta potential of $-10$ mV screened by counterions (determined by dynamic light scattering testing) giving an electric double layer; these counterions ensure efficient dispersion of CNS at a concentration of 40–200 p.p.m. without any aggregation. It is anticipated that the controlled formation of CNS in an aqueous media would take place in hydrogel. PAM/CNS NC gels containing n p.p.m. CNS (named Cn as sample identification) were fabricated by in situ free-radical polymerization of acrylamide (AM) in suspension of CNS. As shown in Fig. 1c for C200 (sample with CNS 200 p.p.m. concentration), the dispersion of the spherulites with diameters $>5$ nm is indeed very homogeneous in the PAM matrix. Because of the low concentration, small size and homogeneous dispersion of the CNS, the transparency of C200 is almost the same as the original PAM hydrogel (Fig. 1d).

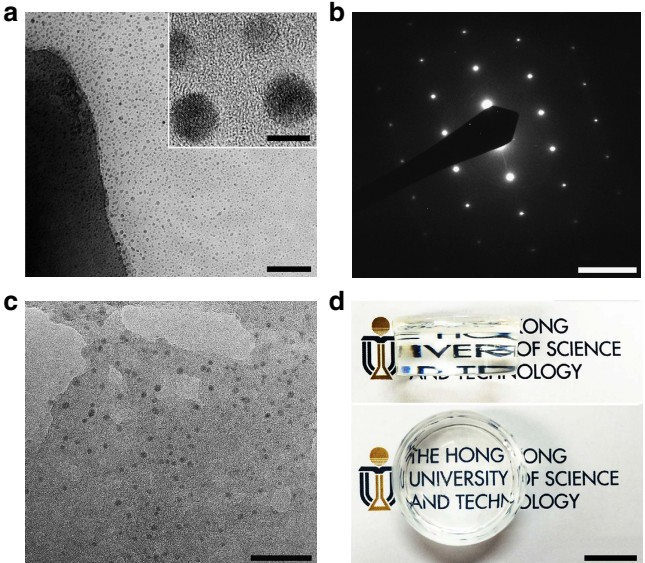

**Figure 1 | PAM enhanced by non-aggregated CNS.** (**a**) TEM image of CNS released from a tricalcium silicate (Ca$_3$SiO$_5$) particle. The scale bars of the whole image and inset are 100 and 5 nm, respectively. (**b**) Electron diffraction patterns of CNS released from a hydrolysed Ca$_3$SiO$_5$ particle. Scale bar, 5 nm$^{-1}$. (**c**) TEM image of CNS dispersed in C200 (PAM/CNS NC gel with 200 p.p.m. CNS concentration). Scale bar, 30 nm. (**d**) Optical photos of a cylindrical C200 specimen with a diameter of 2.7 cm and thickness of 1.2 cm against a background of characters. Scale bar, 1 cm. The transparency of C200 is almost the same as the original PAM hydrogel. PAM represents polyacrylamide; CNS represents calcium hydroxide nano-spherulites.

The link between the PAM functional groups and CNS is established by acid–base reaction between the initiator ammonium peroxydisulfate (APS) and the $Ca(OH)_2$. At room temperature, $S_2O_8^{2-}$ (initiator anion) can be reduced by $OH^-$ in the $Ca_3SiO_5$ suspension, thus polymerization is difficult to be initiated (Supplementary Fig. 1). However, at 0 °C, redox between $S_2O_8^{2-}$ and $OH^-$ is inhibited; the excess $OH^-$ can react with $NH_4^+$ (initiator cation) to form gaseous $NH_3$, which can be removed by a vacuum pump. Moreover, oxygen in the system can also be removed by the pumping process. Therefore, polymerization can start at 0 °C under vacuum conditions (0.01 atm) (Fig. 2). The whole fabrication procedure for the CNS, followed by the PAM/CNS NC gel, is shown in the schematic diagram (Fig. 3).

The connection between CNS and the PAM end group is determined, as being set-up by the ionic bonds between $Ca^{2+}$

and $S_2O_8^{2-}$. To prove this mechanism, time-of-flight secondary ion mass spectrometry (ToF-SIMS) is employed to get the characteristic ion peaks of the original PAM and C200. With the unique PAM backbone ion $CH^-$ as reference, the relative peak areas of the negative ions $SO_4CH^-$ and $SO_4CH_2^-$ are chosen to represent end groups in the PAM chains, while $CN^-$ and $CNO^-$ represent the side groups. Compared with the original PAM, the decreased relative peak intensity of $SO_4CH^-$ and $SO_4CH_2^-$ shows the reduction of the end groups in C200 (Fig. 4a). Similarly, the side group $-CONH_2$ can react with $OH^-$ to form gaseous $NH_3$, which can be promoted by vacuum, and hydrolysed to $-COO^-$, which then connects with $Ca^{2+}$ on the surface of CNS. Such a reaction pathway is evidenced from the decrease of the relative peak intensity of $CN^-$ and $CNO^-$ (Fig. 4a). The appearance of the peaks of $CaSO_4CH^-$ and $CaSO_4CH_2^-$ in C200 strongly supports the chemical bonding between CNS and the PAM end groups (Fig. 4b). Moreover, their relative intensities do not change before and after being stretched for 60 times, indicating the strong connection between CNS and the PAM end group.

**Crosslinked network of PAM/CNS NC gels.** The weight of a piece of freeze-dried xerogel of C40 (sample with CNS 40 p.p.m. concentration), before and after swelling in a large excess of deionized water for 3 weeks, shows that < 0.1 wt% weight is lost, which implies a complete crosslinked network formation of the NC gel with only 40 p.p.m. CNS. At the same volume content and dispensability, the particle number density of the CNS with diameters < 5 nm is ~ 1,000, 2,000, 200 and $5.0 \times 10^5$ times higher than that of hectorite, GO, LDH and titanate(IV) nanosheet, respectively (Supplementary Table 1). The crosslink points disperse much more homogeneously on the surfaces of CNS with the surface area < 78.5 $nm^2$ compared with that of nanosheets with functional surface area > 2800 $nm^2$ (Supplementary Table 1). The distribution and density of the crosslink points are very important in the establishment of the crosslinked network. For example, we obtained $Ca(OH)_2$ micro crystals by cooling a suspension of $Ca(OH)_2$ from room temperature to 0 °C (Supplementary Fig. 2). PAM-based hydrogel named CA200 (sample with $Ca(OH)_2$ micro crystal 200 p.p.m. concentration) was fabricated in the same way as the PAM/CNS

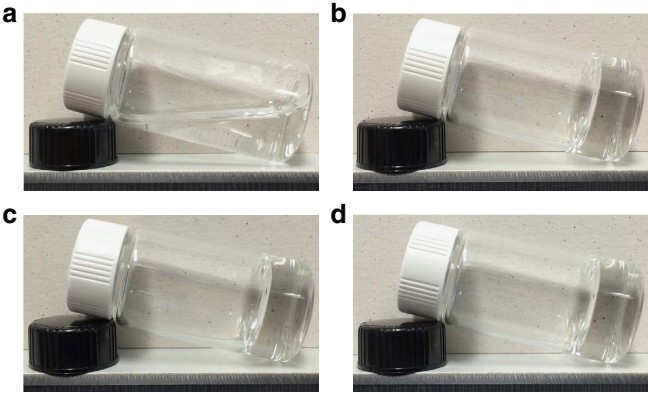

**Figure 2 | Gelation behaviour of PAM/CNS NC gels.** The polymer and tricalcium silicate concentrations were fixed at 20 wt% and 500 p.p.m., respectively. (**a**) After being maintained at 0 °C and 1 atm for 10 days, gelation did not start at 1 atm. (**b**) After being maintained at 0 °C and 1 atm for 1 day, gelation started after 3 h at 0.01 atm. (**c**) After being maintained at 0 °C and 1 atm for 3 days, gelation started after 3 h at 0.01 atm. (**d**) After being maintained at 0 °C and 1 atm for 5 days, gelation started after 3 h at 0.01 atm. PAM/CNS NC gels represent polyacrylamide/calcium hydroxide nano-spherulites nanocomposite hydrogels.

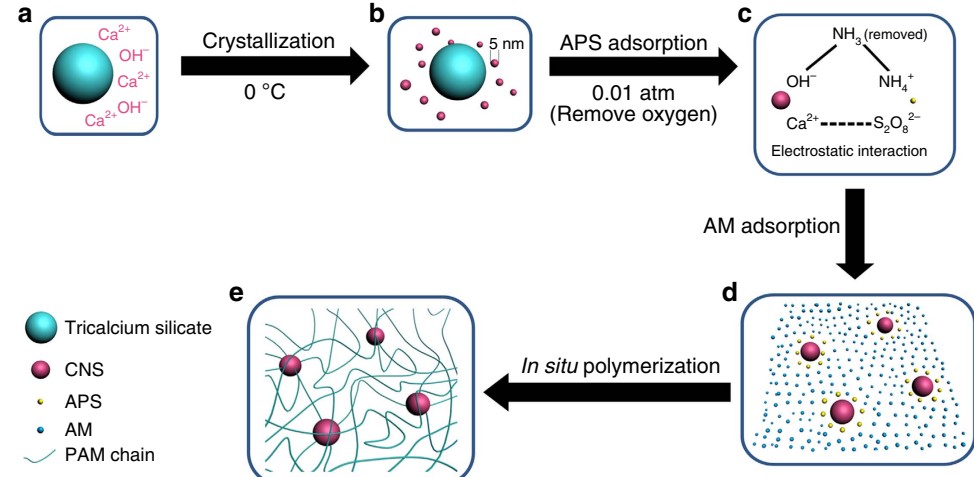

**Figure 3 | Schematic diagram of PAM crosslinked with the CNS released from tricalcium silicate ($Ca_3SiO_5$).** (**a**) $Ca^{2+}$ and $OH^-$ released from the surface of a $Ca_3SiO_5$ particle during the hydration process. (**b**) CNS with diameters < 5 nm formed from the crystallization of calcium hydroxide at 0 °C. (**c**) $S_2O_8^{2-}$ from the initiator APS adsorbed on the surfaces of CNS via electrostatic interaction prompted by low pressure (0.01 atm). (**d**) Introduction of the monomer acrylamide (AM). (**e**) In situ polymerization of PAM on the surfaces of CNS.

NC gel. At the same $Ca(OH)_2$ weight content, the number density of the crosslink points in CA200 is only $1/3.8 \times 10^6$ compared with that in the NC gel (Supplementary Table 1). Such low crosslink density makes a poor network of CA200, which has very limited mechanical enhancement and does not swell, but degenerates and dissolves in water.

Because of the homogeneous distribution and high number density of the CNS particles, the crosslinked network of the NC gel can be effectively controlled by the CNS concentration. Suspension of CNS with concentration 40 p.p.m. (determined by inductively coupled plasma mass spectrometry) can be obtained by dispersing 100 p.p.m. $Ca_3SiO_5$ in water at 0 °C for 3 days. The maximum concentration of CNS is 200 p.p.m., obtained by dispersing 500 p.p.m. $Ca_3SiO_5$ in water at 0 °C for 3 days. This concentration will not increase even with higher $Ca_3SiO_5$ content or longer hydration time, because of the upper limit of calcium cation ($Ca^{2+}$) concentration in $Ca_3SiO_5$ suspension[21]. Figure 5a–f present typical s.e.m. images, showing the morphologies of the freeze-dried samples of C40 and C200 xerogels. Differing from the nanosheet-reinforced NC gels, such as the PAM/LDH NC gel that has a hierarchical porous morphology with an irregular distribution of micro- and nanometre scale pores[16], the freeze-dried xerogel of C40 demonstrates a uniform distribution of porous structures with diameters of 200–400 μm (Fig. 5a). Interconnected nanopores can be observed in the inner walls of the micrometre-size pores (Fig. 5b). Moreover, Fig. 5c further reveals that the diameters of most of these nanopores are <300 nm, indicating the crosslink points disperse homogeneously at the nanometre scale. With an increased crosslinker concentration, the diameters of the porous

structures in the xerogel of C200 are smaller than those of C40, ~20–80 μm, and a great many parallel and crosslinked filamentous structures, with diameters from 50 nm to 2 μm, are distributed regularly inside the pores (Fig. 5d). The obvious embossments at the connection point between the filaments and inner walls of the pores demonstrate the strong connection force (Fig. 5e), which plays an important role in the swelling and mechanical properties of the NC gel.

Compared with C200, the crosslinked network of C40 has a bigger swelling capacity because there are few connecting filaments and larger pore size. With the original shape maintained, the swelling ratio ($Q$) of C40 is 253.4, ~2.6 and 5.7 times higher than that of C100 (sample with CNS 100 p.p.m. concentration, obtained by dispersing 250 p.p.m. $Ca_3SiO_5$ in water at 0 °C for 3 days) and of C200 (Fig. 6a). So, a large range of $Q$ values from 44.8 to 253.4 can be obtained by simply adjusting the concentration of CNS in the NC gels. Unique shish-kebab structures can be observed on the filaments with diameters ~50 nm in the micrometre pores of the C200 xerogels (Fig. 5f and box regions in Fig. 5e). We believe that the bumps with diameters of 100–500 nm are CNS coated with PAM chains. The thickness of the coating layer (100–500 nm) and the distance between the bumps (700 nm–3 μm) suggest that the introduction of CNS has a limited confinement effect on the mobility and flexibility of the polymer molecules for the present NC gels. Another piece of evidence is provided by the glass transition temperature ($T_g$), determined by differential scanning calorimeter measurement (Fig. 6b). Our results show that the

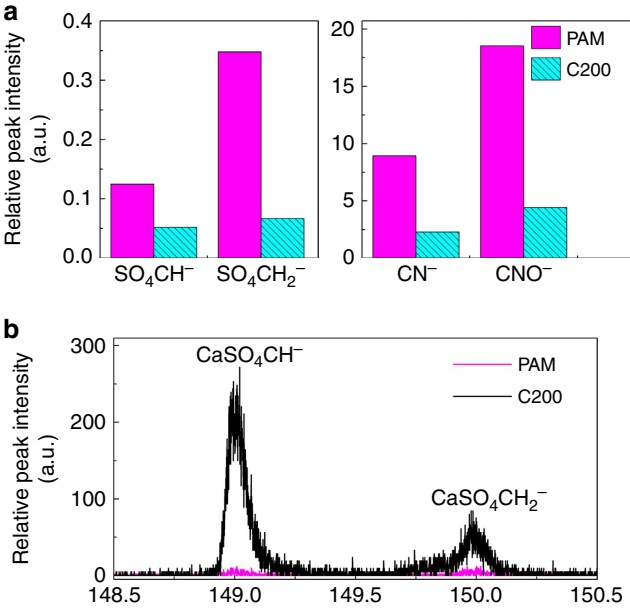

**Figure 4 | ToF-SIMS tests of PAM and C200. (a)** Relative peak intensity of the negative ions representing the end ($SO_4CH^-$ and $SO_4CH_2^-$) and side ($CN^-$ and $CNO^-$) functional groups of the original PAM and C200, with $CH^-$ as the reference. The decreased relative peak intensity of the ions shows the reduction of the end and side groups in C200. **(b)** Peaks of $CaSO_4CH^-$ (mass 149.0) and $CaSO_4CH_2^-$ (mass 150.0) appear in C200, and their relative intensities do not change before and after being stretched 60 times, indicating the strong connection between CNS and PAM end group. C200 represents PAM/CNS NC gel with 200 p.p.m. CNS concentration.

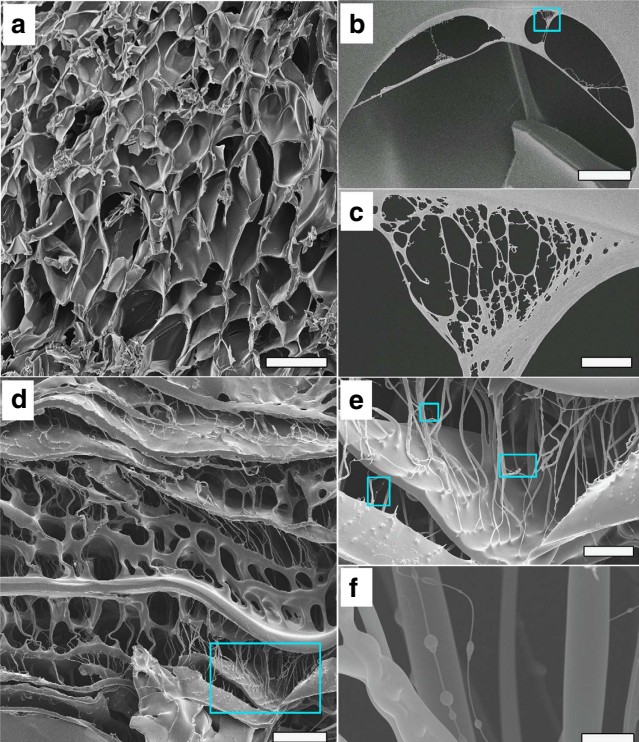

**Figure 5 | Morphology and structure of PAM/CNS xerogels. (a)** Scanning electron microscope (SEM) image of C40 (NC gel with 40 p.p.m. CNS concentration) xerogel. Scale bar, 500 μm. **(b)** Zoomed SEM image of the box region in **a**. Scale bar, 30 μm. **(c)** Zoomed SEM image of the box region in **b**. Scale bar, 2 μm. **(d)** SEM image of C200 (NC gel with 200 p.p.m. CNS concentration) xerogel. Scale bar, 100 μm. **(e)** Zoomed SEM image of the box region in **d**. Scale bar, 30 μm. **(f)** Zoomed SEM image in the region of the top one of the three boxes in **e**. Scale bar, 2 μm.

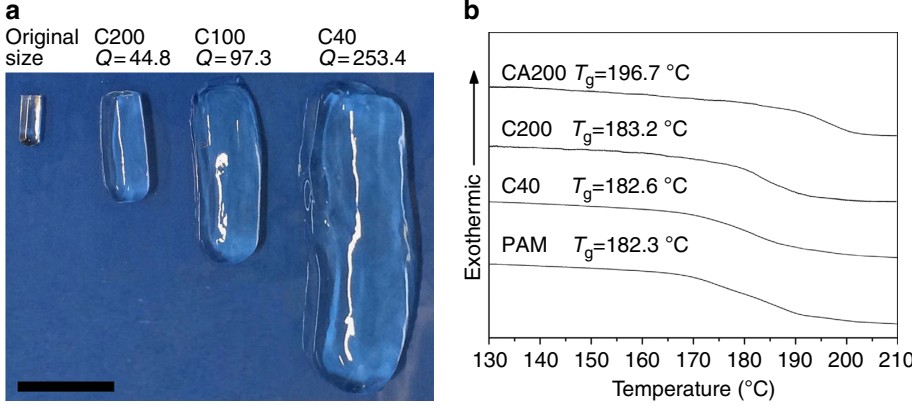

**Figure 6 | Mobility and flexibility of the polymer molecules for PAM/CNS NC gels.** (**a**) An optical photo of swollen hydrogels of C40, C100 and C200 after soaking at a large excess of deionized water for 3 weeks. Scale bar, 60 mm. (**b**) DSC curves showing that the values of $T_g$ for C40 and C200 are slightly higher than that for the neat PAM, and the $T_g$ of CA200 is obviously higher than the others.

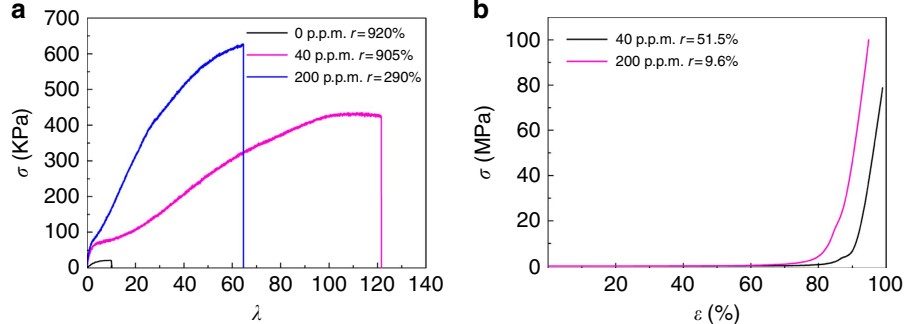

**Figure 7 | Elongation and compressive tests of PAM/CNS NC gels at 25 °C.** (**a**) Representative elongation stress–strain curves of PAM hydrogels containing 0, 40 and 200 p.p.m. CNS. $\sigma$ represents the stress. $\lambda$ represents the stretch ratio. $r$ represents the ratio of the length change along the direction of load after the load is released with respect to the original length of the specimen. (**b**) Compression stress–strain curves of C40 (NC gel with 40 p.p.m. CNS concentration) and C200 (NC gel with 200 p.p.m. CNS concentration). $\varepsilon$ represents the strain.

$T_g$ of the NC gels (C40: 182.6 °C and C200: 183.2 °C) are slightly higher than that of linear PAM (182.3 °C) and almost independent of CNS concentration. In contrast, CA200 has an obvious increased $T_g$ of 196.7 °C, indicating strong restricted force of larger size particles. Therefore, the polymer molecules in the crosslinked network of the NC gels are free, highly flexible and mobile, taking on nearly random conformations between the CNS.

**Mechanical properties of PAM/CNS NC gels.** The mechanical properties of hydrogels are strongly affected by the crosslinked network structure controlled by the concentration of CNS (Fig. 5a–f). To establish the relationship between the network structure and the mechanical properties, we performed elongation tests on the NC gels with different CNS concentrations at room temperature. The representative stress–elongation curves showed that the maximum stress of PAM-based hydrogel increased from 20 to 430 KPa, and the stretch ratio at rupture increased from 10 to 121 (Fig. 7a; Supplementary Movie 1), with the addition of only 40 p.p.m. CNS. The residual deformation ratio ($r$), which is defined as a ratio of the length change along the direction of load after the load is released with respect to the original length of the specimen, can be considered as an index of recoverability. After being stretched 121 times, the specimen C40 could revert to 905% of its initial length, indicating good

recoverability. The stretch ratio at rupture of C40 is higher than the reported highest value for NC gel[10], and the maximum stress is more than four times higher at the same time (c.f. Table 1). The fracture toughness of C40 reaches up to 33.9 MJ m$^{-3}$ because it can achieve high stress and stretch at the same time. The noteworthy tensile properties of C40 can be explained by the following reason: non-aggregated CNS with diameters <5 nm provide homogeneous micro-pores in the crosslinked network, and the lower CNS concentration leads to fewer connecting filaments and larger pore size, which makes for easier deformation (Fig. 5a–c). The stress of C200 reaches 630 KPa at a rupture stretch ratio of 65 and a residual deformation ratio of 290% (Fig. 7a; Supplementary Movie 2), which is ~2.7 times that of the reported highest value of the transparent PAM-based hydrogels with alginate (the stretch ratio is 16 times higher at the same time, c.f. Table 1)[22]. The fracture toughness of C200 is 26.2 MJ m$^{-3}$, which is lower than that of C40, because of the decreased deformation ability of the crosslinked network. However, the smaller pore size and the establishment of the filament structure connecting the inner walls of the pores provide a strong mechanical strength and restoring force for C200 (Fig. 5d–f). In compressive testing, C200 almost reverts to its original size with a residual deformation ratio $r = 9.6\%$, after enduring a compressive stress of 100 MPa at strain of 95% (Fig. 7b; Supplementary Movie 3). In contrast, although C40 can endure a compressive stress of 78 MPa at strain

**Table 1 | Elongation stress and stretch ratio at rupture of synthetic elastic hydrogels with different crosslinkers.**

| Matrix | Particle | Content (p.p.m.) | Stress (KPa) | Stretch ratio |
|---|---|---|---|---|
| Poly(*N*-isopropylacrylamide) | Hectorite[5,6]* | 30,000 | 109 | 8 |
| | | 125,000 | 1,000 | 10 |
| Polyacrylamide | Montmorillonite[10]* | 8,700 | 100 | 118 |
| | | 80,800 | 175 | 85 |
| | Graphene oxide[11]* | 80 | 282 | 31 |
| | | 480 | 385 | 34 |
| | Layered double hydroxide[16] | 8,000 | 34 | 62 |
| | Alginate[22] | 110,000 | 150 | 23 |
| | | 330,000 | 230 | 4 |
| | CNS (this work) | 40 | 430 | 121 |
| | | 200 | 630 | 65 |

CNS, calcium hydroxide nano-spherulites.
*Hydrogels containing high-content clay and graphene oxide have residual and transparency problems.

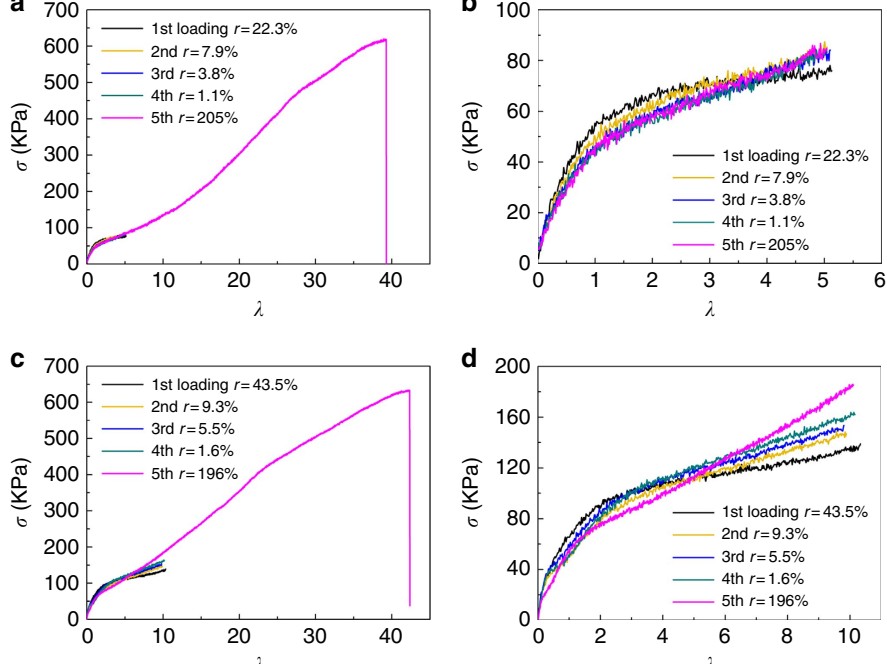

**Figure 8 | Cyclic elongation tests of PAM/CNS NC gels at 25 °C.** (**a**) Cyclic elongation stress–strain curves of C200 (NC gel with 200 p.p.m. CNS concentration) elongated for five-fold in four cycles and elongated to rupture at the fifth cycle. (**b**) Enlarged cyclic elongation stress–strain curves of C200 elongated for five-fold in four cycles. (**c**) Cyclic elongation stress–strain curves of C200 elongated for 10-fold in four cycles and elongated to rupture at the fifth cycle. (**d**) Enlarged cyclic elongation stress–strain curves of C200 elongated for 10-fold in four cycles. $\sigma$ represents the stress. $\lambda$ represents the stretch ratio. $r$ represents the ratio of the length change along the direction of load after the load is released with respect to the original length of the specimen.

of 99% without fracture, it only reverts to about half of its original size with a residual deformation ratio $r = 51.5\%$ (Fig. 7b; Supplementary Movie 4).

The remarkable recoverability of C200 can be further evaluated by cyclic elongation tests (Fig. 8). When stretched five-fold from the first loading to the fifth loading, the cyclic elongation stress–strain curves of C200 nearly overlapped each other, indicating small hysteresis effects in the specimen. The residual deformation ratio $r$ of the first cycle is 22.3%, and further decreases to 1.1% after being stretched over four cycles (Fig. 8a). The stretch ratio of the specimen in each cycle was calculated by the recovered length, with a slightly irreversible deformation after being stretched in the last cycle. So, the stretch ratio at rupture of the C200 specimen in the fifth cycle is ~40 with a residual deformation ratio of 205% due to the previous

irreversible deformation in the pre-cyclic elongation. The enlarged elongation stress–strain curves in Fig. 8b show that the tensile modulus of the first cycle is slightly higher than that of the other cycles when stretched less than four-fold, and decreases with further stretching. It demonstrates that during the initial elongation, there was a slightly irreversible deformation of the crosslinked network. Then, the remarkable recoverability of the NC gel was achieved after the initial conformation adjustment of the crosslinked network. The recoverability can also be observed by stretching the C200 NC gel 10-fold (Fig. 8c,d). Because the NC gel was cyclic stretched with a higher ratio in the first to fourth cycles, the residual deformation ratio $r$ at each cycle was higher than that obtained in the five-fold cyclic tests. However, with a more sufficient conformation adjustment, the residual deformation ratio $r$ in the fifth cycle (stretched to rupture)

decreased to 196%. It demonstrates that good recoverability along the direction of the elongation load can be observed with an initial slight irreversible deformation.

## Discussion

The properties of the PAM/CNS NC hydrogels can thus be tuned through the concentration of CNS, which will give high values for swelling, extensibility, breakage stress and recoverability to the hydrogels. The non-aggregated CNS can also be obtained via a suspension of portland cement instead of $Ca_3SiO_5$ (Supplementary Fig. 3), and can be further applied for property enhancement in other polymeric hydrogels synthesized via *in situ* polymerization. Our results demonstrate that individually separated spherulites with diameters < 5 nm can be obtained from the low-temperature hydration product of $Ca_3SiO_5$, and may help to establish a new NC gel system with enhanced crosslinked networks and properties.

## Methods

**Materials**. AM, APS and $N,N,N',N'$-tetramethyl-ethylenediamine (TEMED; 99%), were purchased from Sigma-Aldrich Chemical Reagent Co., Ltd, and used without further purification. All the chemicals, including calcium carbonate (Alfa Aesar) and silica (4 N, Sinopharm Chemical Reagent Co., Ltd) were of analytical pure reagent (A.R.) grade. In all the experiments, deionized water was used.

**Synthesis of $Ca_3SiO_5$**. Calcium carbonate and silica (molar ratio of 3:1) were first ground to pass through a 63-μm sieve and then evenly mixed for 1–2 h. The obtained fine powder was compressed into pancakes in a lab press and put into a Pt crucible for calcination at 1,500 °C for 5 h, and quickly cooled down to ambient temperature within 10 min. The obtained product was ground into fine powder, compressed and calcined again. After repeating the above process for four cycles, the final $Ca_3SiO_5$ with particle size < 500 nm was obtained (Supplementary Fig. 4).

**Preparation of (CNS) suspension**. $Ca_3SiO_5$ was dispersed in deionized water to obtain a translucent aqueous dispersion, and the system was stirred under ultrasonication for 10 min to yield a homogeneous dispersion. The whole process proceeded at 0 °C in an ice bath.

**Preparation of PAM/CNS NC gels**. The PAM/CNS NC gels were prepared by *in situ* free-radical polymerization. The monomer AM, the initiator APS and the accelerator TEMED were added into the CNS suspension of composition: CNS suspension/AM/APS/TEMED = 60 g/15 g/0.03 g/48 μl. For the complete reaction of CNS and APS, the mixture was kept at 0 °C in an ice bath for at least 72 h. The polymerization process proceeded in a vacuum environment (0.01 atm) at 0 °C.

**Characterization of morphology and structure of the NC gels**. The morphology and distribution of CNS were characterized by transmission electron microscopy (JEOL, JEM-2100, 200 kV), equipped with an energy dispersive spectroscopy system. The structure of the xerogels was characterized by a high-resolution scanning electron microscope (JEOL, model JSM-6700F). The $T_g$ of the hydrogels were determined by a differential scanning calorimeter (TA Q1000). ToF-SIMS was used to demonstrate the existence of the PAM end and side groups, and CNS. Static ToF-SIMS spectra of pure PAM and NC gel C200 were obtained from a ToF-SIMS V spectrometer (ION-TOF GmbH, Münster, Germany). The samples were bombarded with $Bi_1^+$ primary ions, which were accelerated at 25 kV with an average pulsed current of 0.3 pA. The raster area was $200 \times 200$ μm, and the acquisition time for each spectrum was 40 s, corresponding to an ion dose of $< 4 \times 10^{11}$ ions per $cm^2$. Three positive and negative spectra were recorded for each specimen at different locations.

**Mechanical tests**. Mechanical tests were conducted at 25 °C using a MTS (model E44, EXCEED) testing machine. The samples for the elongation tests were of rod-like shape 30 mm in length and 3.2 mm in diameter for the as-prepared PAM/CNS NC gels. The stretched portion of the samples between the two clamps was 2.0 mm for C40 and 1.8 mm for C200, and the samples were elongated at a loading rate of 50 mm min$^{-1}$ (Supplementary Movies 1 and 2). For the compression tests, cylindrical samples of the as-prepared hydrogels were used, with dimensions of 11.8 mm in diameter and 6.1 mm in height for C40, and 10.3 mm in diameter and 5.7 mm in height for C200. The crosshead speed was 1 mm min$^{-1}$, and stopped at the maximum loading of 8,400 N (Supplementary Movie 3 and 4).

**Swelling experiments**. Swelling experiments were performed by immersing the as-prepared hydrogels in a large excess of deionized water at 25 °C for 3 weeks to reach swelling equilibrium. During this period, the water was replaced several times. The swollen samples were then freeze-dried. The swelling ratio ($Q$) was evaluated using $Q = W_s/W_d$, where $W_s$ and $W_d$ are the weights of the swollen sample and the corresponding dried xerogel sample, respectively.

**Data availability**. The data that support the findings of this study are available from the corresponding author upon request.

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

## Acknowledgements

This work was financially supported by the China Ministry of Science and Technology under grant (973 Program: 2015CB655104) managed by HKUST Shenzhen Research Institute and the Hong Kong Research Grants Council under grant 615412.

## Author contributions

G.S. designed and performed all the experiments. Z.L. and R.L. co-designed the experiments. G.S., Z.L. and L.W. analysed the data and wrote the manuscript. R.L. and L.W. supported all the characterizations. L.Z. prepared $Ca_3SiO_5$ with a particle size of <500 nm.

## Additional information

Competing financial interests: The authors declare no competing financial interests.

