## [Peer review file · Nature Communications]

Reviewers' comments:

Reviewer #1 (Remarks to the Author):

The authors developed a new strategy to synthesize calcium hydroxide no-aggregated nano-spherulite (CNS) with diameters smaller than 5 nm, which were used as crosslinker to enhance the polyacrylamide hydrogel. Because the synthesized nanoparticles are so small (~5 nm) compare to conventional nanoparticles (~100 nm in one dimension), they are able to distribute individually and uniformly without any aggregation in the gel solution. As a result, the polyacrylamide/CNS hydrogel is transparent, and show extremely high stretchability, swelling ratio and rupture stress. For example, the polyacrylamide hydrogel containing 400 ppm CNS has a maximum stretch of 121, swelling ratio of 250 and rupture stress of 430 kPa. Consequently, this work provides a new way to improve nanocomposite hydrogel with enhanced crosslinking networks and mechanical properties. I would like to recommend that this paper be published if the authors can address the following issues. z The authors used calcium hydroxide nano-spherulite (CNS) as crosslinker to link the PAAM chains through Ca^{2+} and $-\text{COO}^-$ and SiO_2 via electrostatic interaction. How about CaCl_2 (provide Ca^{2+}) is used as crosslinker instead of CNS? z How does the valance of cations play role? E.g. Al^{3+} . z If polyacrylate is used instead of polyacrylamide, will the authors still expect the polyacrylate/CNS hydrogel to have similar mechanical properties as polyacrylamide/CNS hydrogel? z The schematic in Figure 1(a) contains multiple processes and is unclear. It would be better to separate the multiple processes to each individual process and show the mechanism clearly. z Line 144-145, please explain the reason or provide reference. z Line 200-201 and Figure 3(b), the authors showed when C200 hydrogel is stretched less than 5, the deformation is elastic. Does that mean there is no break of electric interaction in the deformation? However, it is known that Ca^{2+} ionic bonds in alginate are very easy to unzip even at a very small strain. Please provide detailed mechanism for the role that CNS play in the deformation. z Line 203-204 and in Figure 3(b), how about recoverability of gel when stretched larger than 5 (for example, 10)? And also show the corresponding stress and stretch curve in Figure 3(b)

Reviewer #2 (Remarks to the Author):

In this manuscript, the authors reported the preparation of a nanocomposite (NC) hydrogel with very high stretchability by employing calcium hydroxide ($\text{Ca}(\text{OH})_2$) nano-spherulites (CNS, with sub-5nm diameters) as the physical crosslinker. The physical properties such as the mechanical, stretchable and swelling properties were discussed in order to indicate the important effect of no-aggregated CNS on the improved properties of polyacrylamide (PAM) hydrogel. Generally, the study is interesting and recommended for publishing in Nature Communication. However, before published, some crucial issues should be addressed by the authors.

1. In this manuscript, there are several minor written mistakes. Please read the manuscript carefully, and make appropriate corrections and changes. For example, lines 142 and 144, " O°C " should be replaced with " 0°C ".
2. For Figure 1a, the scheme diagram of the hydrogel formation has required some clarifications. It is better to redraw the Figure 1a in order to conveniently observe for the readers.
3. The author used the hydrolysis of tricalcium silicate (Ca_3SiO_5) dispersed in deionized water to obtain the

CNS suspension, and then employed the obtained CNS suspension to prepare the NC hydrogel. I wonder that whether the CNS suspension was purified or not before the preparation of the NC gels. Why?

4. The author claimed that "In aqueous media, CNS have a zeta potential of -10 mV screened by counterions (determined by dynamic light scattering test) to give an electric double layer" (in lines 80-82). Please clarify the concentration of the CNS in the aqueous suspension.

5. Figure 3b should be shown the cyclic elongation stress-strain curves of C200 hydrogel stretched fivefold from first loading to fifth loading. Why did the fifth elongation stress-strain curve stretched near 40 times? Furthermore, it was not clearly observed the cyclic elongation stress-strain curves of C200 hydrogel stretched fivefold. It is better to redraw the Figure 3b in order to conveniently observe for the readers.

6. The author claimed that "When stretched fivefold from first loading to fifth loading, r of C200 decreases from 0.223 to 0.011, indicating a good recoverability of C200, and the recoverability increases by the cycle number of test (Fig. 3b)" (lines 203-205). I wonder why the recoverability increases by the cycle number of test? Moreover, the C200 hydrogel could be stretched up to more than 60 times of their original length. Whether the C200 hydrogel still has good recoverability when it was stretched 10 times?

7. In lines 145-156, the author argued the network structure based on the SEM images of the freeze dried gels. However, the size of the crystal of water is generally larger than the network size. Thus, the SEM images reflect the crystal structures of water. Is it reasonable to characterize the network structure based on the SEM images?

8. In lines 179-184, the author measured the DSC for the fully dried samples, where the PAM was in the glassy state. Is it reasonable to discuss the elasticity of the hydrogel based on the properties in glassy state?

9. In the manuscript, the difference in the mechanical properties between PAM/CNS hydrogel and previously reported carbon dots/PAM, graphene oxide /PAM, clay/PAM NC hydrogels was stated to be due to the presence of CNS. Would the CNS be unique to achieve the excellent mechanical, recoverable and swelling properties of hydrogel (such as the interaction effect between the CNS nanoparticles and the PAM chains) or is it simply due to the relatively smaller size of the CNS embedded in the hydrogel? Several recent studies also reported the addition of functionalized nanoparticles to enhance the mechanical properties of hydrogels, and the authors may compare the NC hydrogels and discuss the possible strategy/implications in material/hydrogel design.

Reviewer #3 (Remarks to the Author):

This manuscript reports the synthesis of no-aggregated calcium hydroxide nano-spherulites and their enhancement to polyacrylamide (PAM) hydrogel. The NC gels show excellent transparency and satisfactory mechanical properties. It may be acceptable after minor revision.

Line 28: I do not agree that "High stress, stretch and recoverability cannot be achieved at the same time in a typical NC gel".

Line 38: it is claimed that "establish a super stretchable and high toughness hydrogel network". But, no toughness values are given.

Please clarify that the value of "stretch" is elongation at rupture, or a stretch ratio between NC and neat

hydrogels. If the stretch is an elongation at rupture, the absolute values of the NC gels are less than reported in the literature, and can be compared with the literature.

The word "Sub-5 nm" looks uncomfortable.

Author response to Reviewer comments

The authors would like to thank the reviewers for their constructive comments and suggestions. Our detailed responses to the specific points raised are given below.

Comments of reviewer 1:

The authors developed a new strategy to synthesize calcium hydroxide non-aggregated nano-spherulite (CNS) with diameters smaller than 5 nm, which were used as crosslinker to enhance the polyacrylamide hydrogel. Because the synthesized nanoparticles are so small (~5 nm) compare to conventional nanoparticles (~100 nm in one dimension), they are able to distribute individually and uniformly without any aggregation in the gel solution. As a result, the polyacrylamide/CNS hydrogel is transparent, and show extremely high stretchability, swelling ratio and rupture stress. For example, the polyacrylamide hydrogel containing 40 ppm CNS has a maximum stretch of 121, swelling ratio of 250 and rupture stress of 430 kPa. Consequently, this work provides a new way to improve nanocomposite hydrogel with enhanced crosslinking networks and mechanical properties. I would like to recommend that this paper be published if the authors can address the following issues.

1. The authors used calcium hydroxide nano-spherulite (CNS) as crosslinker to link the PAAM chains through Ca^{2+} and $-\text{COO}^-$ and $\text{S}_2\text{O}_8^{2-}$ via electrostatic interaction. How about CaCl_2 (provide Ca^{2+}) is used as crosslinker instead of CNS?

Answer: Based on our results, the enhancement of the nanocomposite hydrogels (NC gels) is mainly achieved by the small size of the crosslinker particles. However, we cannot produce similar non-aggregated nano-spherulites of CaCl_2 with diameters less than 5 nm. We have tried to use CaCl_2 to replace CNS in the same condition, but the mechanical properties of the obtained hydrogels are barely improved. Moreover, even for calcium hydroxide itself, if the particle size is not small enough, it cannot be used to enhance the hydrogels. We have reported the findings in lines 136-143 in the revised manuscript and Supplementary Fig. S2 in the revised supplementary information.

2. How does the valance of cations play role? E.g. Al^{3+} .

Answer: The valance of cations in the crosslinker particles may influence the electrostatic interaction between the particles and polymer chains. Further research is needed to identify such interesting mechanism. If we can

find a suitable initiator and reaction condition (appropriate temperature or pressure) to establish the connections between the Al^{3+} and polymer chains, we may use nanoparticles containing Al^{3+} to enhance the hydrogels.

3. If polyacrylate is used instead of polyacrylamide, will the authors still expect the polyacrylate/CNS hydrogel to have similar mechanical properties as polyacrylamide/CNS hydrogel?

Answer: The authors appreciate the suggestion by the reviewer. Actually we have already started extended work in applying the CNS in other hydrogel systems such as polyacrylate, poly(N-isopropylacrylamide), polyvinyl alcohol and so on. We will report the relevant work in the near future.

4. The schematic in Figure 1(a) contains multiple processes and is unclear. It would be better to separate the multiple processes to each individual process and show the mechanism clearly.

Answer: The authors agree to the suggestion by the reviewer. We have redrawn the schematic and separated the multiple processes, which can be seen in Figure 3 in the revised manuscript.

5. Line 144-145, please explain the reason or provide reference.

Answer: The reason and the relevant reference have been added in lines 150-152 in the revised manuscript. The maximum concentration of CNS from our method is 200 ppm. Because the CNS are formed by the crystallization of calcium cation (Ca^{2+}) released from tricalcium silicate, and there is an up limit of Ca^{2+} concentration in tricalcium silicate suspension, the concentration of CNS has an up limit too.

6. Line 200-201 and Figure 3(b), the authors showed when C200 hydrogel is stretched less than 5, the deformation is elastic. Does that mean there is no break of electric interaction in the deformation? However, it is known that Ca^{2+} ionic bonds in alginate are very easy to unzip even at a very small strain. Please provide detailed mechanism for the role that CNS play in the deformation.

Answer:

The mechanism for the role that CNS play in the deformation can be found in the description of the ToF-SIMS results (lines 111-124 and Figure 4 in the revised manuscript). We infer that the connection between the CNS and PAM end group is set up by the ionic bonds between Ca^{2+} and $\text{S}_2\text{O}_8^{2-}$, which is believed to be stronger than the electric interaction in the alginate. We added ToF-SIMS tests on C200 before and after stretching (Fig. 4b of the revised manuscript), which both show the appearance of CaSO_4CH^- and $\text{CaSO}_4\text{CH}_2^-$ characteristic ions in C200. The relative intensity of these two unique ions did not change before and after stretching, which partially proved our mechanism.

7. Line 203-204 and in Figure 3(b), how about recoverability of gel when stretched larger than 5 (for example, 10)? And also show the corresponding stress and stretch curve in Figure 3(b).

Answer: The cyclic elongation tests of C200 stretched tenfold have been completed based on the reviewer's suggestion. The relevant results and discussion can be found in lines 231-237 and Figure 8 in the revised manuscript.

Comments of reviewer 2:

In this manuscript, the authors reported the preparation of a nanocomposite (NC) hydrogel with very high stretchability by employing calcium hydroxide (Ca(OH)₂) nano-spherulites (CNS, with sub-5nm diameters) as the physical crosslinker. The physical properties such as the mechanical, stretchable and swelling properties were discussed in order to indicate the important effect of non-aggregated CNS on the improved properties of polyacrylamide (PAM) hydrogel. Generally, the study is interesting and recommended for publishing in Nature Communication. However, before published, some crucial issues should be addressed by the authors.

1. In this manuscript, there are several minor written mistakes. Please read the manuscript carefully, and make appropriate corrections and changes. For example, lines 142 and 144, "0{degree sign}C" should be replaced with "0 {degree sign}C".

Answer: The authors apologize for the written mistakes. The revised manuscript has been checked carefully and the relevant mistakes mentioned by the reviewer are corrected in lines 148 and 150 in the revised manuscript.

2. For Figure 1a, the scheme diagram of the hydrogel formation has required some clarifications. It is better to redraw the Figure 1a in order to conveniently observe for the readers.

Answer: The authors appreciate for the suggestion by the reviewer. We have redrawn the schematic and added detailed clarifications in the Figure caption, which can be seen in Figure 3 in the revised manuscript.

3. The author used the hydrolysis of tricalcium silicate (Ca₃SiO₅) dispersed in deionized water to obtain the CNS suspension, and then employed the obtained CNS suspension to prepare the NC hydrogel. I wonder that whether the CNS suspension was purified or not before the preparation of the NC gels. Why?

Answer: The CNS suspension was not purified before the preparation of the NC gels. The stability of the suspensions (100 to 500 ppm) of tricalcium silicate (Ca₃SiO₅) with a size around 500 nm is not very good. The Ca₃SiO₅ particles precipitated slowly during the hydrolysis process. After hydrolysis for 3 days, most of the

Ca_3SiO_5 particles precipitated at the bottom of the vial, remaining a transparent suspension of CNS. In this case, the particles of Ca_3SiO_5 in the CNS suspension are very difficult to find in the TEM test (We only found few Ca_3SiO_5 particles in a copper grid of the TEM sample, one of which is shown in Figure 1a in the revised manuscript). So, we believe that such a small amount of the remaining Ca_3SiO_5 particles hardly influence the properties of the NC gels, and the CNS suspension can be used without purification.

4. The author claimed that "In aqueous media, CNS have a zeta potential of -10 mV screened by counterions (determined by dynamic light scattering test) to give an electric double layer" (in lines 80-82). Please clarify the concentration of the CNS in the aqueous suspension.

Answer: The concentration of the CNS in the aqueous suspension with a zeta potential of -10 mV is 200 ppm, and has been clarified in line 90 in the revised manuscript.

5. Figure 3b should be shown the cyclic elongation stress-strain curves of C200 hydrogel stretched fivefold from first loading to fifth loading. Why did the fifth elongation stress-strain curve stretched near 40 times? Furthermore, it was not clearly observed the cyclic elongation stress-strain curves of C200 hydrogel stretched fivefold. It is better to redraw the Figure 3b in order to conveniently observe for the readers.

Answer: The previous description of the cyclic tests has been extended in lines 218-237 in the revised manuscript. The first to fourth elongation stress-strain curves show the recoverability of the C200 specimen at the same stretch ratio. The fifth elongation stress-strain curve is intended to prove that after previous cyclic elongation, the C200 can still reach a similar stress and stretch ratio at rupture with that of the original elongation test. The stretch ratio of the specimen in each cycle was calculated by the recovered length with a slightly irreversible deformation after stretching in the last cycle. So, the stretch ratio at rupture of the C200 specimen in the fifth cycle is about 40 due to the previous irreversible deformation in the pre-cyclic elongation.

The authors appreciate for the suggestions by the reviewer. Figure 3b has been extended to become Figure 8 in the revised manuscript. Enlarged figures of the cyclic elongation stress-strain curves of C200 hydrogel stretched fivefold and tenfold have been added. Actually, we had thought of adding the recovering curves to make the circular curves for each cycle, but the recoverability of the C200 specimen is very good, which makes the circular curves have some overlapped part that is difficult to be distinguished. So, we present the cyclic elongation stress-strain curves in a similar way to that in Kamata's report (Science, 2014, vol 343, page 873-875).

6. The author claimed that "When stretched fivefold from first loading to fifth loading, r of C200 decreases from 0.223 to 0.011, indicating a good recoverability of C200, and the recoverability increases by the cycle number of test (Fig. 3b)" (lines 203-205). I wonder why the recoverability increases by the cycle number of

test? Moreover, the C200 hydrogel could be stretched up to more than 60 times of their original length. Whether the C200 hydrogel still has good recoverability when it was stretched 10 times?

Answer: The explanation for the increased recoverability by the cycle number of the tests can be found in lines 228-231 in the revised manuscript. During the initial elongation, there was an irreversible deformation caused by the conformation adjustment of the crosslinked network. The conformation adjustment of the crosslinked network is decreased by the cycle number of test, so the recoverability increases.

The C200 hydrogel still has good recoverability when stretched 10 times. The relevant tests and description are added in lines 231-237 and Figure 8 (c) and (d) in the revised manuscript.

7. In lines 145-156, the author argued the network structure based on the SEM images of the freeze dried gels. However, the size of the crystal of water is generally larger than the network size. Thus, the SEM images reflect the crystal structures of water. Is it reasonable to characterize the network structure based on the SEM images?

Answer: The freeze dried gels for the SEM tests were obtained from thoroughly swelled hydrogels, in which the crosslinked network of the hydrogel was fully expanded and could not be changed during the freeze-drying process. Thus it is reasonable to characterize the network structure based on the SEM images of the freeze dried gels. It is a quite common method for the characterization of the network structure and has been used by many research groups for hydrogels, for example Xia et al.(Nature Communications, 2013, vol 4, doi:10.1038/ncomms3226).

8. In lines 179-184, the author measured the DSC for the fully dried samples, where the PAM was in the glassy state. Is it reasonable to discuss the elasticity of the hydrogel based on the properties in glassy state?

Answer: The mobility of the polymer chain is restricted by the filler particles. Normally, the restriction is much stronger in the glassy state than that in the hydrogel state. In our work, we compared the restriction effect of different samples in the glassy state. We found that the CNS have a very slight restriction effect on the mobility of the polymer chains in the glassy state, so the restriction effect should be even weaker in the hydrogel. This method has been widely used to characterize the elasticity of hydrogels, for example in Hu's report (Advanced Materials, 2014, vol 26, page 5950-5956).

9. In the manuscript, the difference in the mechanical properties between PAM/CNS hydrogel and previously reported carbon dots/PAM, graphene oxide /PAM, clay/PAM NC hydrogels was stated to be due to the presence of CNS. Would the CNS be unique to achieve the excellent mechanical, recoverable and swelling properties of hydrogel (such as the interaction effect between the CNS nanoparticles and the PAM chains) or is it simply due to the relatively smaller size of the CNS embedded in the hydrogel? Several recent

studies also reported the addition of functionalized nanoparticles to enhance the mechanical properties of hydrogels, and the authors may compare the NC hydrogels and discuss the possible strategy/implications in material/hydrogel design.

Answer: We believe that the remarkable mechanical enhancement of PAM/CNS hydrogel is mainly due to the relatively smaller size of the CNS embedded in the hydrogel. If the size of the CNS is increased, it will not be able to enhance the hydrogel. We have proved this mechanism by calcium hydroxide micro crystals as described in lines 136-143 in the revised manuscript and Supplementary Fig. S2 in the revised supplementary information. To our knowledge, there are no crosslinker nanoparticles with size smaller than the CNS, so the mechanical properties of the NC gels enhanced by CNS are better than others.

Of course, we cannot rule out the effect of the interaction between CNS and PAM. At this stage, it is difficult to quantify this effect and certainly necessitate further study in future through some specially designed experiments, e.g. keeping particle size while changing the surface functionalities.

Comments of reviewer 3:

This manuscript reports the synthesis of non-aggregated calcium hydroxide nano-spherulites and their enhancement to polyacrylamide (PAM) hydrogel. The NC gels show excellent transparency and satisfactory mechanical properties. It may be acceptable after minor revision.

1. Line 28: I do not agree that "High stress, stretch and recoverability cannot be achieved at the same time in a typical NC gel".

Answer: We agree and the sentence has been changed to "high stress, stretch and recoverability are difficult to achieve at the same time in a typical NC gel" in lines 56-57 in the revised manuscript.

2. Line 38: it is claimed that "establish a super stretchable and high toughness hydrogel network". But, no toughness values are given.

Answer: The authors appreciate the suggestion by the reviewer. The elongation toughness of the NC gels C40 and C200 has been calculated with the value of 33.9 and 26.2 MJ m⁻³, respectively. The relevant description can be found in lines 200 and 208 in the revised manuscript.

3. Please clarify that the value of "stretch" is elongation at rupture, or a stretch ratio between NC and neat hydrogels. If the stretch is an elongation at rupture, the absolute values of the NC gels are less than reported in the literature, and can be compared with the literature.

Answer: Actually, the value of "stretch" is the elongation at rupture. The authors appreciate the comment by the reviewer, and we have searched the literature more thoroughly. Before that, we thought that the highest value was 62 times reported by Hu et.al. in *Advanced Materials*, 2014, vol 26, page 5950-5956. After a careful search, we found the reported highest value of the stretch ratio at rupture was 118 times (11800% in the literature), as reported by Gao et.al. in *ACS Applied Materials & Interfaces*, 2015, vol 7, page 5029–5037. This work used exfoliated montmorillonite (MMT) layers as the crosslinker, achieving good mechanical and especially good healing properties, and has now been cited and compared in our revised manuscript. However, the 118 times stretch is still a slightly lower than our results (121 times). At a similar stretch ratio, the stress and toughness of our sample (C40) are about 4 and 3 times higher than that reported by Gao et.al., while the transparency and recoverability of our samples are much better. To our knowledge, 118 times is the highest stretch ratio that we can find, and we welcome the reviewer to provide us relevant literatures if there are any higher values.

4. The word "Sub-5 nm" looks uncomfortable.

Answer: We agree, and all the words "Sub-5 nm" have been changed to "less than 5 nm" in the revised manuscript.

REVIEWERS' COMMENTS:

Reviewer #1 (Remarks to the Author):

The authors have answered all the comments that the reviewer gave, and added detailed description and figures to support their points. The revised version is concise and complete. No further revision is needed.

Reviewer #2 (Remarks to the Author):

This reviewer is satisfied with the authors' response. Publish as it.